# Efficacy of the Continuous Resuscitation Training with the Gap Period Due to the COVID-19 Pandemic

Que N. N. Tran *, Takeshi Moriguchi , Norikazu Harii, Junko Goto, Daiki Harada, Hisanori Sugawara, Junichi Takamino, Masateru Ueno, Hirobumi Ise, Akino Watanabe, Hiroki Sakata, Kengo Kondo, Natsuhiko Myose and Fuki Sakurabayashi

Emergency & Critical Care Medicine Department, Graduate School of Medicine, Faculty of Medicine, University of Yamanashi, 1110 Shimokato, Chuo City, Yamanashi 409-3898, Japan; tmoriguchi@yamanashi.ac.jp (T.M.); nharii@yamanashi.ac.jp (N.H.)
* Correspondence: g19ddm19@yamanashi.ac.jp; Tel.: +81-55-273-9812

**Abstract:** (1) Objective: This study evaluates the effects of simulation education at our institute on cardiac arrest resuscitation regarding knowledge, attitude, and practice (KAP) over a five-year period (2016–2020). (2) Subjects: Staff responded to the annual survey questionnaires followed by monthly training in Basic Life Support/Advanced Cardiovascular Life Support (BLS/ACLS) and Immediate Cardiac Life Support (ICLS) of the Japanese Association for Acute Medicine (JAAM) in 2016, 2017, and 2018. Additionally, in-house training was implemented in 2019 without post-assessment followed by training suspension in 2020 due to the COVID-19 pandemic. The last delivery of the survey questionnaires was in late 2020 for KAP retention measurement. (3) Measurements and Results: The self-efficacy level of BLS/ACLS/ICLS KAP of the survey respondents was analyzed using a five-point Likert scale. The mean self-efficacy level of BLS/ACLS/ICLS KAP increased over time, and that of the trained people was three-fold that of the untrained people. Trainees that had no cardiopulmonary resuscitation (CPR) experience gained the BLS/ACLS/ICLS key-point self-efficacy level, which we call the "Grip 14" in this study, as high as their untrained counterparts who had three-time CPR experience. Training suspension lessened the BLS/ACLS/ICLS KAP in both groups. (4) Conclusions: Continuous training enhances not only the BLS/ACLS/ICLS KAP of trainees but also of their untrained colleagues. The training likely had the same efficacy as the CPR experience.

**Keywords:** arrhythmia; cardiopulmonary resuscitation/CPR; medical education; efficiency; self-assessment



## 1. Introduction

Basic life support (BLS) education is an appropriate choice for hospitals to enhance and qualify the fundamental skills of bystanders for resuscitation [1–3]. In addition to BLS courses, advanced cardiovascular life support (ACLS) courses from the American Heart Association (AHA) guidelines based on the International Liaison Committee on Resuscitation 2015 [4,5] progressively improve the practices of automated and manual defibrillation, airway management, and related pharmacology. Furthermore, such advanced courses are designed to help learners with early recognition and management of cardiopulmonary emergencies and stroke [4,5]. Frequent AHA updates show the effectiveness of BLS/ACLS training, especially with revised courses, including in-hospital skills [6,7]. In its most recent recommendations, the International Liaison Committee on Resuscitation 2020 also highlighted the role of team simulation training [8]. Likewise, Immediate Cardiac Life Support (ICLS) courses of the Japanese Association for Acute Medicine (JAAM) are used and certified nationwide in Japan [9]. In particular, ICLS aims to equip learners with relatively similar goals to ACLS, apart from the management of peri-arrest arrhythmias and stroke. Stated differently, ICLS focuses on advanced skills in clinical with respect to related-cardiopulmonary arrest, such as AED (automated external defibrillator), airway

management, and medication administration [9]. There are also certain knowledge achievements in BLS education in Japan for both medical [10] and non-medical staff [11]. However, the effects of combined BLS/ACLS/ICLS training in Japan remain unknown.

In putting BLS training to practice, the efficacy of training-course modification [12,13] and the factors of knowledge, attitude, and practice (KAP) retention have been revealed [14–16]. First, simulation education is efficient in both public and healthcare settings [12,13]. Second, although clinical experiences appear to be a strong factor that can skip retraining courses [16], aging, on the contrary, negatively affects both the attitude towards performing BLS [14] and skill-retention capability [15]. Lastly, because our survey was conducted before and during the COVID-19 pandemic, it is likely that the impact of the pandemic can be observed as an objective factor affecting the attitudes of lay rescuers towards BLS/ACLS/ICLS training [17].

"Self-efficacy," a concept defined first by Bandura, can possibly provide a groundwork for bystanders' to perform BLS/ACLS/ICLS education practically [18]. On the one hand, by applying his social cognitive theory (SCT) to "explain and predict psychological changes" [18,19], the self-efficacy level of survey respondents might quantify their confidence in the BLS/ACLS/ICLS performance and their attitude towards the subsequent courses. Stated differently, self-efficacy level is used not only as a tool to evaluate the training effectiveness but can also be used as a quantitative tool for enhancing the practicality of training in our hospital in the future. On the other hand, there are four factors affecting human behaviors or their beliefs: performance accomplishments, vicarious experience, verbal persuasion, and emotional arousal [18], which can lead to changes in the self-efficacy level of people. Thus, it is necessary to observe and evaluate the respondents' self-efficacy level in their milieus.

The primary aim of this study was to assess the self-efficacy levels of survey respondents after introducing BLS/ACLS/ICLS training in our institute. The secondary objective was to evaluate the impact of COVID-19 on KAP retention. Having said that, there were many biases in the study, such as the heterogeneous experience in clinical practice with respect to the diversity of the study population, the complexity of the pandemic's impacts, the dynamics of survey respondents over years, etc. However, because there has been no study revealing the BLS/ACLS/ICLS KAP self-efficacy level of training attendants and non-attendants before and after the COVID-19 pandemic, the current study could give an insight of the effectiveness of the training as well as the effects of the pandemic.

## 2. Materials and Methods

The study was reviewed and the need for approval was waived by our Institutional Review Board in the University of Yamanashi Hospital. The details are as below:

Chairman: Professor Yamagata Zentaro.
Approval number: 1598.
Approval date: 11th January 2017.
Study title: The study of the efficacy of cardiopulmonary resuscitation training
Informed consent was waived because the study was just questionary.

Procedures were followed in accordance with the ethical standards of the responsible Committee on Human Experimentation (institutional or regional) and the Helsinki Declaration of 1975.

- Study design: cross sectional studies.
- Setting: single-center study.
- Interventions: None.

Between 2016 and 2020, our institute implemented monthly BLS/ACLS/ICLS courses in 2016, 2017, 2018, and 2019. We also delivered post-assessment questionnaires to all hospital personnel every March of the year following the completion of the annual training program. The results of the questionnaires were collected within two weeks after the delivery. The study was divided into two main stages (Scheme 1) because of the training gap in 2020 that occurred due to the COVID-19 pandemic. Additionally, in the pre-pandemic

era, the training program was conducted in a classroom format with instructor-led, hands-on practice. In an ordinary class, the trainers were usually emergency or ICU doctors who had received the corresponding certificate, and the trainees were the staff working in either medical or non-medical environments who had registered for their attendance.

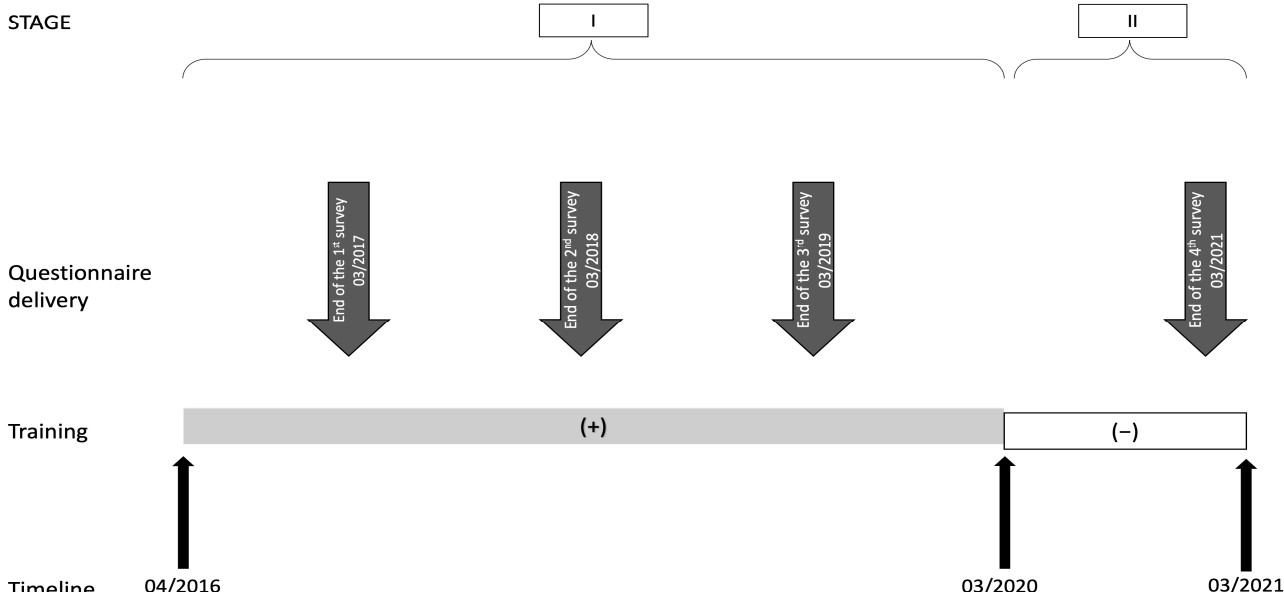

**Scheme 1.** Timeline of the BLS/ACLS/ICLS training program and post-assessment. Scheme 1 shows the timeline of the BLS/ACLS/ICLS training program and the post-assessment from April 2016 to March 2021. It was divided into two main stages, with a gap in training during 2020. Stage I was marked by the introduction of the in-house training program in 2016, followed by a post-training survey in the same year until the routine training courses and their post-assessment occurred in 2017, 2018, and 2019, only skipping the post-assessment in 2019. Stage II included the delivery of survey questionnaires at the end of 2020 after skipping the training in 2020. In other words, the self-efficacy levels of the participants in 2016, 2017, 2018, and 2020 were scored as the post-assessment of the training in 2016, 2017, 2018, and 2019.

We originally developed a survey questionnaire that was structured in the same format during the study (see Supplementary Figure S1). The first twenty questions were scored on a 5-point Likert scale, with a maximum score of 2 points and a minimum score of −2 points. While the maximum score illustrated the strong agreement of a survey respondent towards a specific parameter, the minimum point showed a strong disagreement, and a "zero" expressed a neutral perspective. The validation of the questionnaire showed the excellent internal consistency and excellent integrity of the 20 items, with Cronbach's alpha higher than 0.9 (Supplementary Table S1).

Moreover, while the first twenty questions generally measured the self-efficacy levels of BLS/ACLS/ICLS KAP, we developed a group of fourteen in-depth CPR/AED items (questions 1, 4, 5, 6, 7, 8, 9, 10, 11, 12, 13, 14, 18, and 20) named the "Grip 14" in order to assess the extent to which BLS/ACLS/ICLS key-point was achieved in the survey population. The "Grip 14" self-efficacy level in relation to the frequency of CPR experiences was analyzed in order to clarify the practicality of training. Based on the self-assessment of the survey respondents, we calculated the mean and the standard deviation of the BLS/ACLS/ICLS KAP, or the first twenty questions, as well as the BLS/ACLS/ICLS key-point, or the "Grip 14". After dividing the population into two groups, namely, trained and untrained people in terms of experiencing the BLS/ACLS/ICLS training in general, these groups were statistically compared based on the self-efficacy level of BLS/ACLS/ICLS KAP and key-point. The reason why the parameters of "Grip 14" were picked up and grouped was to score the performance of the lay rescuers that were initiating resuscitation

by highlighting the most essential in-depth CPR/AED skills. Thus, while the average self-efficacy level of the first twenty parameters can show the BLS/ACLS/ICLS KAP retention after training, the "Grip 14", on the other hand, provides an evaluation of the training practicality in correlation with the real CPR/AED experiences of the lay rescuers or the bystanders. As a consequence, the scores of KAP retention and practicality can help us to evaluate the efficacy of this training program.

The statistics used in this study were the two-proportion z-test and the two-sample *t*-test assuming equal variances.

## 3. Results

Table 1 shows the demographic characteristics of the participants during the five-year period from 2016 to 2020. The number of female participants was almost twice that of their male counterparts, and the average age group of the participants was the 40s. Table 2 shows how many people participated in the training courses that were held in our institute in the 2016–2018 period as well as the rate of post-assessment respondents in the present study.

**Table 1.** Demographic characteristics of survey cohorts.

| Characteristics | 2016 | 2017 | 2018 | 2020 |
|---|---|---|---|---|
| Sex | | | | |
| Male | 599 (34.97%) | 477 (31.51%) | 521 (33%) | 582 (33.74%) |
| Female | 1114 (65.03%) | 1037 (68.49%) | 1058 (67%) | 1143 (66.26%) |
| Total | 1713 | 1514 | 1579 | 1725 |
| Age | | | | |
| On average (years old) | 38.64 | 38.66 | 47.02 | 37.66 |
| Organizational position | | | | |
| Administration | 165 (10.12%) | 164 (11.35%) | 167 (10.98%) | 177 (10.62%) |
| Personnel | 1466 (89.88%) | 1281 (88.65%) | 1354 (89.02%) | 1489 (89.38%) |
| Total | 1631 | 1445 | 1521 | 1666 |
| Work environment | | | | |
| Clinical | 1276 (74.88%) | 1154 (76.93%) | 1193 (75.99%) | 1454 (84.93%) |
| Non-clinical | 428 (25.12%) | 346 (23.07%) | 377 (24.01%) | 258 (15.07%) |
| Total | 1704 | 1500 | 1570 | 1712 |
| Experience in clinical procedures performance | | | | |
| Yes | 931 (54.76%) | 855 (56.77%) | 971 (61.65%) | 1082 (62.94%) |
| No | 769 (45.24%) | 651 (43.23%) | 604 (38.35%) | 637 (37.06%) |
| Total | 1700 | 1506 | 1575 | 1719 |

**Table 2.** Numbers of BLS/ACLS/ICLS trainees and survey respondents.

| Numbers of People | 2016 | 2017 | 2018 | 2019–2020 * |
|---|---|---|---|---|
| University Hospital Staff | 2028 | 2099 | 2027 | 2089 & 1991 |
| Participants of BLS/ACLS/ICLS training | 193 (9.52%) | 126 (6.00%) | 119 (5.87%) | 34 (1.63%) |
| Survey respondents | 1731 (85.36%) | 1526 (72.70%) | 1590 (78.44%) | 1741 (87.44%) |

* The in-house training was implemented in 2019 without post-assessment, followed by training suspension in 2020 due to COVID-19. The delivery of survey questionnaires was in late 2020 for KAP retention measurement.

Figure 1 shows a significant increase in the percentage of people with BLS/ACLS/ICLS training experiences over time—from more than 45% in 2016 to more than 60% in 2020 (Cochran–Armitage test, z = −8.40, *p* < 0.001). In particular, the percentage of trained people significantly increased in Stage I, and it maintained sustainability in Stage II (z = −1.05, *p* = 0.15). The upper-tailed two-sample z-test results during 2016–2017 and 2017–2018 were z = −4.98, *p* < 0.001 and z = −2.07, *p* = 0.02, respectively.

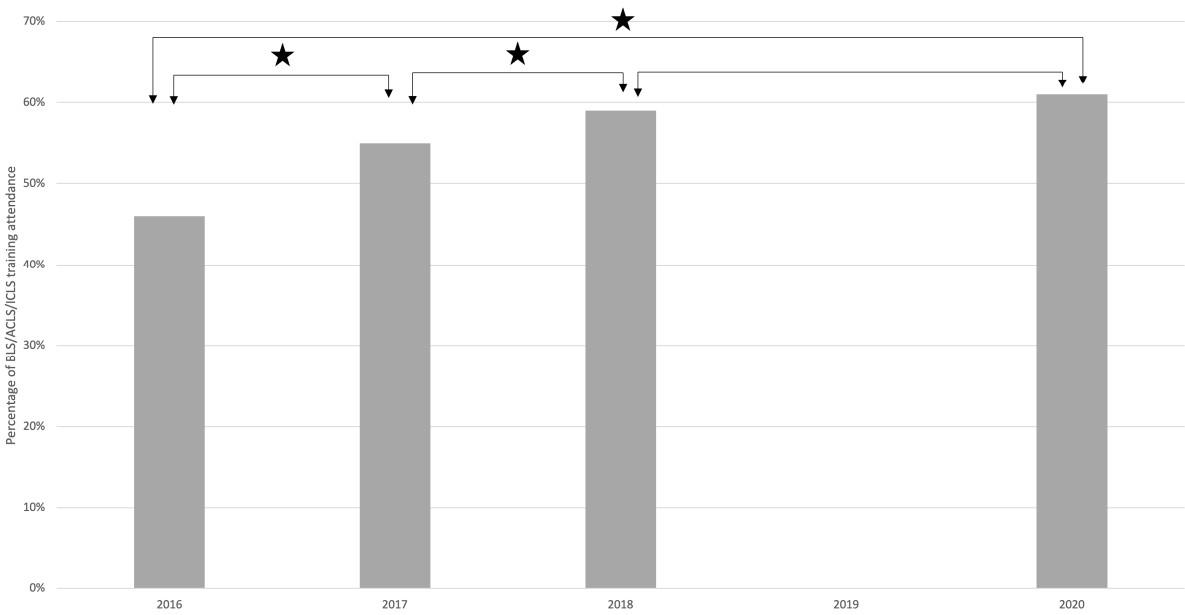

**Figure 1.** Percentage of BLS/ACLS/ICLS training experience. This figure shows the rate of people who experienced the BLS/ACLS/ICLS training courses. The black star-shaped symbols represent significant increases in the percentage of trained people over time (a = 0.05). The *p* values for 2016–2017, 2017–2018, 2018–2020, and 2016–2020 are *p* < 0.001, *p* = 0.02, *p* = 0.15, and *p* < 0.001, respectively.

Figure 2 shows the statistical differences in BLS/ACLS/ICLS KAP self-efficacy levels over time in both the trained and the untrained people. The upper-tailed *t*-test results were represented by black star symbols for the trained people and black crescent symbols for the untrained people. Likewise, the white star and white crescent symbols represented the lower-tailed *t*-test results. The scores on average for the 20 questions in the training attendance group increased significantly in the five-year period of 2016–2020 despite a significant decrease in Stage II, in which the populations were as follows: 2016 (*M* = 1.32; *SD* = 0.53), 2017 (*M* = 1.35; *SD* = 0.51), 2018 (*M* = 1.41; *SD* = 0.48), and 2020 (*M* = 1.35; *SD* = 0.51). The statistics were analyzed using the upper-tailed two-sample *t*-test for the periods of 2016–2017, 2017–2018, and 2016–2020, resulting in $t_{32236} = -6.09$, *p* < 0.001; $t_{34031} = -10.05$, *p* < 0.001, and $t_{36023} = -6.56$, *p* < 0.001, respectively, and the lower-tailed *t*-test was analyzed in the 2018–2020 period ($t_{37818} = -10.45$, *p* < 0.001). However, similar differences could be observed in their counterparts, with remarkably lower average scores: 2016 (*M* = 0.06; *SD* = 0.92), 2017 (*M* = 0.21; *SD* = 0.89), 2018 (*M* = 0.34; *SD* = 0.88), and 2020 (M = 0.31; *SD* = 0.86). The one-tailed *t*-test results corresponded to those of the training-attendance group: $t_{29688} = -14.07$, *p* < 0.0001; $t_{23810} = -11.69$, *p* < 0.001; $t_{29727} = -23.70$, *p* < 0.001; and $t_{23849} = -3.15$, and *p* = 0.001, respectively.

Figure 3 shows significantly higher self-efficacy levels for "Grip 14" among the trained people by the frequency of witnessing emergency settings, which ranged from never (*M* = 1.15; *SD* = 1.17), once (*M* = 1.39; *SD* = 0.97), twice (*M* = 1.53; *SD* = 0.83), and thrice (*M* = 1.54; *SD* = 0.84) to many times (*M* = 1.73; *SD* = 0.68). This compared to the group of untrained people, where the frequency ranged from never (*M* = −0.10; *SD* = 1.47), once (*M* = 0.54; *SD* = 1.46), twice (*M* = 0.96; *SD* = 1.17), and thrice (*M* = 1.12; *SD* = 1.18) to many times (*M* = 1.30; *SD* = 1.09). The lower-tailed two-sample *t*-test results were represented

by black star symbols, and those, by increasing frequency, were $t_{48952} = 99.40$, $p < 0.001$, $t_{8300} = 31.62$, $p < 0.001$, $t_{5011} = 19.03$, $p < 0.001$, $t_{3818} = 10.50$, $p < 0.001$, and $t_{20002} = 28.60$, $p < 0.001$, respectively.

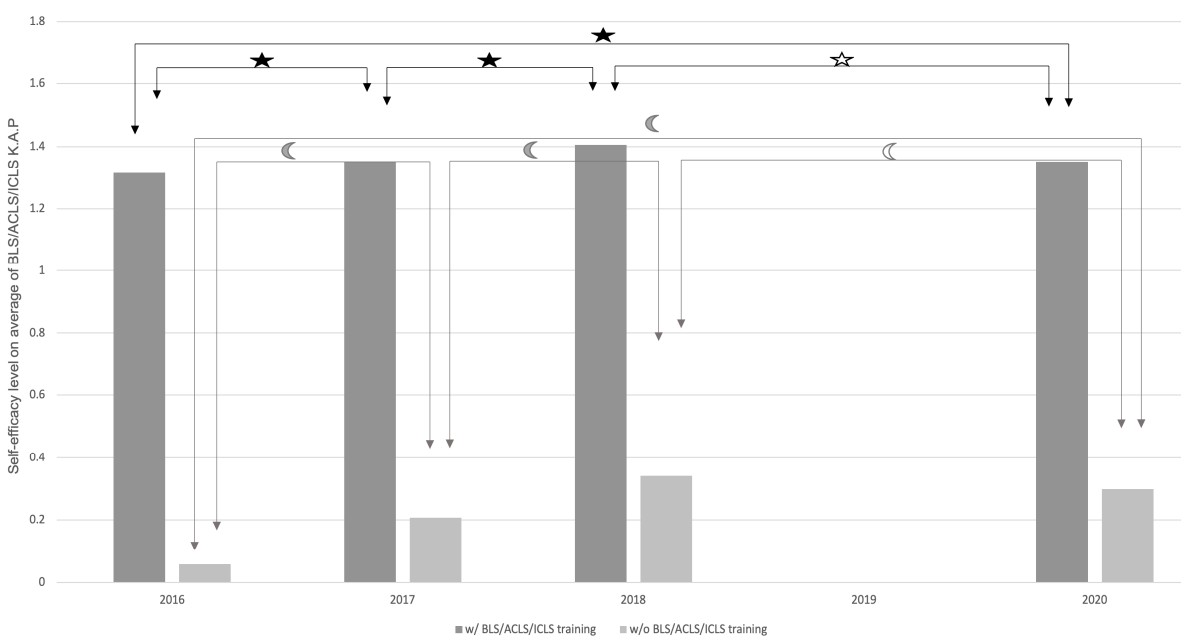

**Figure 2.** Self-efficacy level on average of BLS/ACLS/ICLS KAP with and without BLS/ACLS/ICLS training attendance. The survey respondents were separated into training attendants (the star-shaped group) and non-attendants (the crescent-shaped group) based on their participation in BLS/ACLS/ICLS training. The black symbols illustrate significant increases in the self-efficacy levels of 20 parameters on average, while the white symbols symbolize significant decreases (a = 0.05). Most of the *p* values are <0.001, except for the *p*-value of the white crescent-shaped symbol in Stage II (0.001).

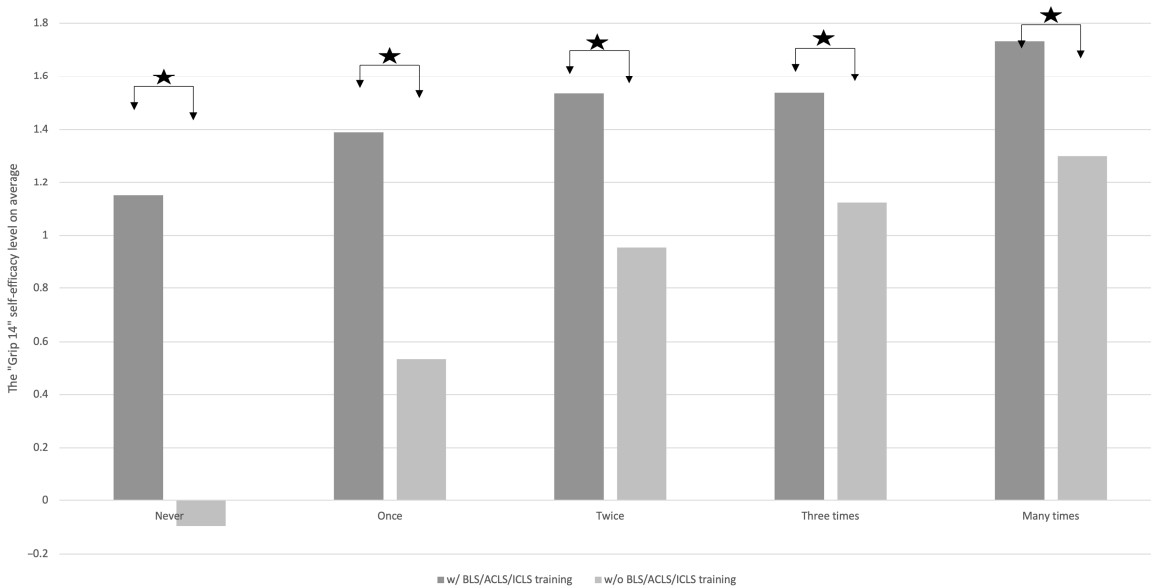

**Figure 3.** The "Grip 14" average self-efficacy levels by frequency of witnessing BLS/ACLS/ICLS-requiring emergency settings. This figure emphasizes the differences in the self-efficacy levels of the training attendants and the non-attendants after a certain frequency of witnessing emergency settings requiring BLS/ACLS/ICLS skills. The black star-shaped symbols represent significantly higher self-efficacy levels of training attendants (a = 0.05). All *p* values are <0.001.

## 4. Discussion

An in-depth examination of intensive healthcare in the clinical environment showed that more than 50% of the survey population had experienced clinical procedures (Table 1). Moreover, when focusing on Stage II, "a hidden peak" in BLS/ACLS/ICLS KAP after training in 2019 could be anticipated (Figure 2).

The first consideration was to evaluate the training efficacy, which included two components: the percentage of training experience and the self-efficacy level of the trainees. While the percentage of trainees who attended annually accounted for less than 10%, the reason for the declining trend may have been a decrease in motivation of the trainees for the next year's training after acquiring some certain KAP from their prior attendance, and the post-assessment rates were over 70% (Table 2). This revealed a higher percentage of people who had experienced BLS/ACLS/ICLS training, possibly before they worked for the institute (Table 2). Notably, as shown in Figure 1, there was a 15% significant increase in trained people after five years (from 46% in 2016 to 61% in 2020), regardless of the absence of training courses in 2020, particularly in the earlier period of study, or Stage I (Scheme 1). Because of the gap in BLS/ACLS/ICLS training in 2020 due to the COVID-19 pandemic, a fall in the self-efficacy levels of knowledge could be predicted in the latter period of the survey (Figure 2). Nevertheless, an in-depth understanding of Figure 2 shows that there was a significant increase in the self-efficacy levels of BLS/ACLS/ICLS KAP of both the trained and the untrained people in the five-year period in general, with a three-fold higher level of the trained people compared to the untrained people. This finding proves the values of training courses, as has been mentioned in the literature worldwide [1–8] and also, specifically, in Japan [10]. However, because of the use of a post-assessment viewpoint in comparison, (for example, the self-efficacy levels of bystanders, but neither the survival rate of patients [1] nor the proportion of patients achieving a return of spontaneous circulation [3]), the training efficacy in this study is less relevant to the survival rate than that of prior studies.

The second consideration is to include CPR experience in relation to the training role. Figure 3 illustrates the trend of "the more frequent the practice, the more confident the feeling," which emphasizes the role of clinical experience in the literature [8,16]. Nonetheless, as shown in Figure 3, in terms of a grasp of BLS/ACLS/ICLS key-point—or what we have called "Grip 14"—trained people with no CPR experience had a similar self-efficacy level to untrained people with only three-time CPR experience. In conclusion, while the important role of CPR experience in practice cannot be denied, the present study demonstrates the benefit of simulation education by unveiling the higher self-efficacy level of trained individuals with less experience. This finding also relates to the literature [8], but it is from a different perspective, which is based on the comparison of self-efficacy levels. Moreover, although Schmitz et al. showed in their study [16] that experience can be a strong factor in skipping training, this study illustrates that training maintains a decisive role in terms of BLS/ACLS/ICLS key-point retention.

From the two abovementioned considerations, namely, a three-fold higher self-efficacy level of trainees and two similar mean self-efficacy levels, whether of trainees with no experience or untrained colleagues with only three-time CPR experience, the training likely has the same efficacy as the CPR experience. Having said that, this study had several confounding factors that interacted with the KAP of the survey population in complex ways. First, the impact of the COVID-19 pandemic was an objective factor that could lessen the self-efficacy levels of BLS/ACLS/ICLS KAP due to the lack of training (Figure 2). On the other hand, it was likely a trigger for a sudden significant increase in the participants' eager attitude toward BLS/ACLS/ICLS training by the 17th question analysis in Stage II (Supplementary Figure S1). This finding shares the same phenomenon as Birkun's study [17]. Furthermore, it cannot be denied that the observation of in-real clinical settings regarding whether or not the BLS/ACLS/ICLS training improved cardiac arrest during the pandemic would be challenging to answer due to many factors, such as the crisis of ICU beds, the risk of exposure to rescuers, etc., which are mentioned in the literature [20]. Hence,

although the current study has to some extent revealed the motivated attitude of general people toward the training, it brings us back to the cornerstone method we used regarding the definition of their self-efficiency level in relation to the process of human behavior changes [18]. Second, collective behavior was a variable that could create an increase in the self-efficacy level of untrained individuals by chance, especially in a harmony-oriented Asian society in Japan with better listening skills [21]. Actually, this behavior can be more clearly revealed by the observation of two parallel trends of the self-efficacy levels of BLS/ACLS/ICLS KAP of trained and untrained people in Figure 2. Indeed, despite the fact that the untrained group received no direct benefits from the in-house training, their mean self-efficacy level increased in the earlier period and decreased in the latter period corresponding to that of their trained colleagues, while this should have kept its sustainability as a control group unless this confounder got involved in the scenario. In short, the pandemic had two-way influences on BLS/ACLS/ICLS KAP, and collective behavior may have affected the motivation for training courses, and this could open the possibility of biases and confounding factors in the study.

When it comes to continuous training in the post-pandemic era, there are some strong points that make the benefits outweigh the risks. First, along with the shift to virtual worlds, the online portion of the BLS/ACLS/ICLS training should be optimal without diminishing the role of hands-on skills. Second, the resuscitation scenario for COVID-19 and other highly contagious disease-suspected patients should be designed for future classes, focusing on adequate personal protective equipment (PPE) supply for both the victims and the rescue team [20]. Lastly, the follow-up should be ensured for both of the related parties.

The research has its limitations. First, the dynamics and circumstances in the five-year study shown in Scheme 1 resulted in the inadequacy of the baseline information before 2016 and that of the regular training in 2020 (Stage II). As a substitute, the post-assessment in FY 2016–2017 was taken into consideration as the baseline for the efficacy evaluation of the five-year training. Second, as mentioned earlier in the Section 1, there were various areas of expertise in the survey population that could be the bias in the evaluation of the BLS/ACLS/ICLS self-efficacy level with respect to clinical experience. Lastly, the KAP of survey respondents were measured by their self-assessment rather than their practical skills. Hence, the sufficient evaluation of continuous training by both practical skills and post-assessment can improve the study in the future.

## 5. Conclusions

Enhancing BLS/ACLS/ICLS KAP through continuous training not only benefits trained attendants but also their untrained colleagues. The training likely had the same efficacy as the CPR experience in our study. Herein, attending a single session of BLS training was observed to yield an equivalent impact on BLS/ACLS/ICLS key-point as undergoing thrice the practical encounters with CPR. In contrast, the disruption of training decreased KAP retention. Meanwhile, the COVID-19 pandemic caused the motivated attitude of people toward resuscitation training, but it also posed new challenges to implementing the training in the post-pandemic era.

**Supplementary Materials:** The following supporting information can be downloaded at: https://www.mdpi.com/article/10.3390/ime2030018/s1, Figure S1: Self-efficacy level of BLS/ACLS/ICLS knowledge, attitude, and practice (KAP); Table S1: The internal consistency and integrity of the BLS/ACLS/ICLS KAP items in the survey questionnaire.

**Author Contributions:** Q.N.N.T. and T.M. analyzed and interpreted the survey questionnaires regarding the BLS/ACLS/ICLS KAP. Q.N.N.T. and T.M. performed the analyses. Q.N.N.T., T.M., N.H., J.G., D.H., H.S. (Hisanori Sugawara), J.T., M.U., H.I., A.W., H.S. (Hiroki Sakata), K.K., N.M. and F.S. conducted the investigation. T.M. and N.H. provided supervision and revision. Q.N.N.T. was a major contributor in writing the manuscript. All authors have read and agreed to the published version of the manuscript.

**Funding:** This research received no external funding.

**Institutional Review Board Statement:** The study was conducted in accordance with the Declaration of Helsinki, and approved by the Institutional Review Board (or Ethics Committee) of the University of Yamanashi Hospital (protocol code 1598 and approval date of 11 January 2017). Ethical review and approval were waived for this study due to the questionary design of the study.

**Informed Consent Statement:** Not applicable.

**Data Availability Statement:** Available in the Japanese Database Network at https://center6.umin.ac.jp/cgi-open-bin/ctr_e/ctr_view.cgi?recptno=R000058309.

**Conflicts of Interest:** The authors declare that there is no competing interest to declare.

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
