# Peer review of "Efficacy of the Continuous Resuscitation Training with the Gap Period Due to the COVID-19 Pandemic"

_ime, doi:10.3390/ime2030018_

Round 1
Reviewer 1 Report
The original manuscript entitled 'Efficacy of the continuous resuscitation training with the gap period due to the COVID-19 pandemic' describes the time trends of continuous BLS/ACLS/ICLS KAP training intervened by the COVID-19 pandemic era. The scientific motif is interesting, and the message is important for emergency medical education. However, this reviewer has several concerns regarding this manuscript.
1. This is a single-center study, and this center is considered the University of Yamanashi Hospital isn't it? This reviewer noticed that by reading the supplementary file. This center should be clarified in the main text.
2. This study is based on the questionnaire composed of Q1-Q20 regarding the self-efficacy to assess SLA/ACLA/ICLS KAP. Is this original or cited (literature?)? The questionnaire study requires the internal consistency and integrity of all the questions included. How is Cronbach's alpha?
3. With respect to the statistical analysis, comparisons between the data of contiguous years were repeated in Figures 1 & 2. Such multiple comparisons sometimes mislead the overall trend spanning 2016 to 2020. How are the results obtained by other statistical methods such as the Mantel-Heanszel test or the Cochran-Armitage test for trends?
4. COVID-19 has a significant impact on overall medical education. In the Discussion, the authors described that the COVID-19 pandemic has two-way influences on the BLA/ACLS/ICLS KAP. With this respect, authors should cite brand-new information on cardiac arrest resuscitation in the COVID-19 era, if any literature or guideline exists.
Author Response
The original manuscript entitled 'Efficacy of the continuous resuscitation training with the gap period due to the COVID-19 pandemic' describes the time trends of continuous BLS/ACLS/ICLS KAP training intervened during the COVID-19 pandemic era. The scientific motif is interesting, and the message is important for emergency medical education. However, this reviewer has several concerns regarding this manuscript.
- This is a single-centre study, and this centre is considered the University of Yamanashi Hospital isn't it? This reviewer noticed that by reading the supplementary file. This centre should be clarified in the main text.
- We add this part in the Method section accordingly.
- This study is based on the questionnaire composed of Q1-Q20 regarding the self-efficacy to assess SLA/ACLA/ICLS KAP. Is this original or cited (literature?)? The questionnaire study requires the internal consistency and integrity of all the questions included. How is Cronbach's alpha?
- We add Supplementary Table 1 accordingly.
- With respect to the statistical analysis, comparisons between the data of contiguous years were repeated in Figures 1 & 2. Such multiple comparisons sometimes mislead the overall trend spanning 2016 to 2020. How are the results obtained by other statistical methods such as the Mantel-Heanszel test or the Cochran-Armitage test for trends?
- We used the Cochran-Armitage test for Figure 1, and the Student test for Figure 2.
- COVID-19 has a significant impact on overall medical education. In the Discussion, the authors described the COVID-19 pandemic has two-way influences on the BLA/ACLS/ICLS KAP. With this respect, authors should cite brand-new information on cardiac arrest resuscitation in the COVID-19 era, if any literature or guideline exists.
- We add this part in the Discussion section accordingly.
Reviewer 2 Report
The scope of the introduction should be expanded.
The originality of the Manuscript should be better emphasized.
Materials and Methods: Sufficient.
It is sufficient to interpret the results found.
The discussion is not enough. The result found in the study should be interpreted more clearly.
What does the manuscript suggest? The authors should emphasize.
Minor editing of English language required
Author Response
The scope of the introduction should be expanded.
The originality of the Manuscript should be better emphasized.
- We add this part in the introduction section accordingly.
Materials and Methods: Sufficient.
It is sufficient to interpret the results found.
The discussion is not enough. The result found in the study should be interpreted more clearly.
What does the manuscript suggest? The authors should emphasize.
- We add this part in the Discussion and Conclusion sections accordingly.
Reviewer 3 Report
Thank you for the opportunity to peer review. This paper was an interesting study on the relationship between training and self-efficacy. However, many biases exist, including coronaviruses, clinical experience, and areas of expertise. This study was not adjusted.
I think a major modification of the study design is necessary to evaluate the aims of this study. For that reason, I'm sorry to say, it would be hard to get published.
The following is a minor point.
Intro
#1 The explanation of self-efficiency is insufficient. Furthermore, the significance of verifying the relationship between training and self-efficacy were not presented.
#2 The description of what is not known from the prior literature regarding what you are trying to clarify in this study is insufficient.
materials
#3 Is this questionnaire valid? Is it original? There is no mention of reliability or validity assessment.
#4 There is no mention of bias.
Results
#5 The part from L123 should be mentioned in the discussion part, because this part is author's opinion.
Author Response
Thank you for the opportunity to peer review. This paper was an interesting study on the relationship between training and self-efficacy. However, many biases exist, including coronaviruses, clinical experience, and areas of expertise. This study was not adjusted.
I think a major modification of the study design is necessary to evaluate the aims of this study. For that reason, I'm sorry to say, it would be hard to get published.
- Thank you for your review. We understand they are the limitations of the current study. We hope to have better study designs for future research in the post-pandemic era.
The following is a minor point.
Intro
#1 The explanation of self-efficiency is insufficient. Furthermore, the significance of verifying the relationship between training and self-efficacy was not presented.
- We add this part in the Introduction section accordingly.
#2 The description of what is not known from the prior literature regarding what you are trying to clarify in this study is insufficient.
- We add this part in the Introduction section accordingly.
materials
#3 Is this questionnaire valid? Is it original? There is no mention of reliability or validity assessment.
- We add Supplementary Table 1 accordingly.
#4 There is no mention of bias.
- We add this part in the Introduction section accordingly.
Results
#5 The part from L123 should be mentioned in the discussion part because this part is the author's opinion.
We put this part in the Discussion section accordingly.
Round 2
Reviewer 3 Report
Thank you for the revised version according to my peer review.
I have a concern regarding the questionnaire. There is an additional statement regarding reliability, but there is no mention of whether this questionnaire has been previously validated or was originally developed. You should state that point clearly and sufficiently.
Also, you should state if Grip14 was developed by you for this study or it is a citation from other prior literature.
Author Response
Thank you for your review.
I revised the manuscript followed by the two points you stated. I cited them here for your reference. Likewise, you can find more details in the Method section.
1. There is an additional statement regarding reliability, but there is no mention of whether this questionnaire has been previously validated or was originally developed. You should state that point clearly and sufficiently.
=> "We originally developed a survey questionnaire that was structured in the same format during the study (Supplementary Figure 1). The first twenty questions were scored on a five-point Likert scale, with a maximum score of 2 points, and a minimum score was -2 points. While the maximum score illustrated the strong agreement of a survey respondent toward a specific parameter, the minimum point showed a strong disagreement, and a “zero” expressed a neutral perspective. The validation of the questionnaire showed the excellent internal consistency and integrity of those 20 items with Cronbach’s alpha higher than 0.9 (Supplementary Table 1)."
2. Also, you should state if Grip14 was developed by you for this study or it is a citation from other prior literature.
=> "Moreover, while the first twenty questions generally measured the self-efficacy levels of BLS/ACLS/ICLS KAP, we developed a group of fourteen in-depth CPR/AED items (questions 1, 4, 5, 6, 7, 8, 9, 10, 11, 12, 13, 14, 18, and 20) named the “Grip 14” in order to assess the extent to which BLS/ACLS/ICLS key-point was achieved in the survey population."